

# Investigating the impact of atmospheric stability on thunderstorm outflow winds and turbulence

Patrick Hawbecker[1], Sukanta Basu[2], and Lance Manuel[3]

[1]Department of Marine, Earth, and Atmospheric Sciences, North Carolina State University, Raleigh, NC 27695, USA.
[2]Faculty of Civil Engineering and Geosciences, Delft University of Technology, Delft, the Netherlands
[3]Department of Civil, Architectural, and Environmental Engineering, University of Texas at Austin, Austin, Texas 78712, USA.

*Correspondence to:* Sukanta Basu (s.basu@tudelft.nl)

**Abstract.**

Downburst events initialized at various hours during the evening transition (ET) period are simulated to determine the effects of ambient stability on the outflow of downburst winds. The simulations are performed using a pseudo-spectral large eddy simulation model at high resolution to capture both the large-scale flow and turbulence characteristics of downburst winds.

First, a simulation of the ET is performed to generate realistic initial and boundary conditions for the subsequent downburst simulations. At each hour in the ET, an ensemble of downburst simulations is initialized separately from the ET simulation in which an elevated cooling source within the model domain generates negatively buoyant air to mimic downburst formation.

The simulations show that while the stability regime changes, the ensemble mean of the peak wind speed remains fairly constant (between 35–38 m s$^{-1}$) and occurs at the lowest model level for each simulation. However, there is a slight increase in

intensity and decrease in the spread of the maximum outflow winds as stability increases as well as an increase in the duration in which these strongest winds persist. This appears to be due to the enhanced maintenance of the ring vortex that results from the low-level temperature inversion, increased ambient shear, and a lack of turbulence within the stable cases. Coherent turbulent kinetic energy and wavelet spectral analysis generally show increased energy in the convective cases and that energy increases across all scales as the downburst passes.

*Copyright statement.*

# 1   Introduction

Severe thunderstorms producing tornadoes and extreme winds have garnered a lot of attention in the wind engineering and wind energy communities due to the high risk of structural damage to buildings and wind turbines caused by these events. In 2016 alone, severe thunderstorms caused an estimated $14 billion USD in insured losses within the United States; roughly 60%

of the total estimated insured losses from natural catastrophes for the year (Source: 2017 Munich Re as of February 2017). Historically, there have been several instances of structures directly impacted by these events resulting in studies focused on





estimating the loads generated by these winds on structures such as buildings (Chen, 2008; Sengupta et al., 2008) and electricity transmission towers (Holmes and Oliver, 2000; Oliver et al., 2000; Savory et al., 2001; Chay et al., 2006). With recent growth of the wind energy industry, the occurrence of severe winds impacting wind turbines has become more common. For example, footage of a tornado moving through a wind farm in Harper County, Kansas which occurred in 2012 shows the blades of

a wind turbine being stripped away and projected (source: https://www.youtube.com/watch?v=Egdtlnv6Gio), while a severe thunderstorm event that struck the Buffalo Ridge Wind Farm in 2011 ended up causing several wind turbines to lose their blades and the buckling of a tower (Hawbecker et al., 2017). Traditionally, surveys of damage to features such as buildings, crops, and trees are conducted after an event and, based on the severity of the damage, a rating of 1–5 is assigned based on the Enhanced-Fujita scale (McDonald and Mehta, 2006), or the EF Scale; formerly the Fujita Scale (Fujita, 1971, 1981).

The EF Scale uses damage-wind relationships and bases the ratings on observed damage and design codes. However, indirect estimation of extreme winds based on damage surveys and damage-wind relationships are known to be imprecise and uncertain at times (Reynolds, 1971; Doswell III and Bosart, 2001; Marshall, 2002; Doswell III et al., 2009; Edwards et al., 2013); still, due to the dearth of high-density wind measurement networks, the wind engineering community is compelled to use these estimations. In recent years, high-resolution numerical modeling is gaining momentum as a viable alternative that can serve as

a surrogate to in-situ data (Dahl et al., 2017).

While high-resolution observations of near-surface winds for severe wind events are rare, climatologies of severe wind events have been developed via several intensive field campaigns and by utilizing storm reports from the National Weather Service (NWS). One such climatology of thunderstorms producing severe winds over the contiguous United States was developed about three decades ago by Kelly et al. (1985). This climatology is generated based on NWS storm reports of thunderstorm-related

wind damage over a 29-year period from 1955 through 1983 but it is impossible to distinguish the storm type (i.e., downburst, derecho, rear flank downdraft, etc.) from these reports. Around the same time, Wakimoto (1985) generated another storm climatology based solely on observed downbursts from the Joint Airport Weather Studies (JAWS) project (McCarthy et al., 1982). Even though the size of the analyzed dataset was much smaller than that of Kelly et al. (1985), Wakimoto (1985) was able to document the circadian traits of downbursts. Figure 1a shows a regenerated version of the data from Figure 6 in

Kelly et al. (1985) while Figure 1b shows the regenerated data from Figure 4 in Wakimoto (1985). Here, it can be seen that severe wind events occur most often during the late afternoon. As the sun begins to set, the number of severe wind events begins to sharply decrease. However, late evening and even nocturnal events, while less common, do occur for all wind events (Figure 1a) and downbursts (Figure 1b). The period of the day roughly two hours before sunset to two hours after sunset is known as the *evening transition* (ET) when, in a matter of hours, the atmospheric boundary layer (ABL) transitions from

the daytime convective boundary layer (CBL) to the nighttime stable boundary layer (SBL). The CBL is characterized by large turbulent eddies spanning the depth of the boundary layer generated from the heating of the surface. As the sun sets and the surface begins to cool, the SBL develops where a surface-based layer of stably stratified air forms along with an increase in low-level vertical wind shear. Further, due to the lack of surface heating, the turbulence from large convective eddies is greatly reduced (Stull, 1988). The neutral boundary layer (NBL) exists only very briefly between the convective and

stable regimes (Park et al., 2014) where vertical sensible heat flux near the surface is zero and there is an adiabatic lapse rate



throughout the boundary layer (Stull, 1988). At the beginning of the ET, when the boundary layer is still in a convective state, solar heating is able to generate convective instabilities in the boundary layer that can initiate thunderstorms (Wakimoto, 2001). However, as the sun sets, this convective instability source is lost, thus, decreasing the number of such severe events.

Another interesting observation can be made from Figure 1a: as the atmospheric stability increases during the ET, the percentage of events considered strong or violent remains relatively constant. In other words, atmospheric stability does not seem to play a role in modulating the ratio of severe to non-severe wind events. This counter-intuitive result should be studied in detail. In fact, the impact of increased stability on tornadogenesis has been investigated recently; the climatologies of tornadic events have suggested a peak in the occurrence frequency of tornadoes in the early evening (Coffer and Parker, 2015). In the present study, we will probe further into how stability impacts downburst winds and into such implications on the wind energy and wind engineering communities. Other extreme wind events (such as derechos, rear flank downdrafts, etc.) are out of scope of the present study.

Downbursts, defined as such by Fujita (1985), produce low-level diverging outflow capable of generating severe winds. Such outflow is dominated primarily by the horizontal components of the wind field; however, they are enhanced by strong *vertical* motions at low levels as well. Downbursts are typically associated with thunderstorm activity and are commonly separated into two categories: *dry* and *wet* (Wakimoto, 1985). Dry downbursts are associated with little or no rain at the surface during the outflow winds. On the other hand, wet downbursts are associated with rainfall at the surface during the time of the outflow winds. In order to understand distinctions between a downburst classified as dry or wet and the potential impact of atmospheric stability on downburst winds, basic concepts of downburst formation are explained here. The two main drivers to generate a downburst are hydrometeor loading (i.e., drag from falling hydrometeors) to initiate the sinking motion, and latent cooling from melting, evaporation, and sublimation of the hydrometeors (Wakimoto, 2001). These produce a pocket of cold (relative to the environment), negatively buoyant air that descends towards the surface and spreads out laterally as it reaches the ground. With that basic picture in mind, several factors (e.g. the ambient turbulent motions of the CBL, or the surface inversion with increased low-level wind shear in the SBL) play a role in modifying the downburst winds. For example, in a typical CBL or NBL regime, the pocket of air would remain negatively buoyant throughout its descent to the surface. However, in the SBL where a low-level inversion is present, the downburst air could become less negatively buoyant (or even positively buoyant) as it descends through the low-level stable layer (Proctor, 1989). Since the resulting horizontal wind speeds are a function of the maximum downdraft velocity, $w_{min}$, if $w_{min}$ is smaller (larger), then the outflow wind speeds will in turn be smaller (larger) (Proctor, 1989; Mason et al., 2009). Thus, if the downdraft encounters a highly turbulent environment, the mixing and entrainment of ambient environmental air can weaken the negative buoyancy of the parcel and, in turn, decrease the outflow wind speeds.

For several decades, the wind engineering and the atmospheric science communities have studied downburst winds. Due to a lack of high-resolution low-level observations during these events, as previously mentioned, numerical models as well as laboratory experiments (e.g., Lundgren et al., 1992; Yao and Lundgren, 1996; Sengupta and Sarkar, 2008; Zhang et al., 2013; Jesson et al., 2015) have been utilized to study downburst dynamics and wind fields. Numerical models employed include





analytical models, Reynolds-Averaged Navier-Stokes (RANS) models, large eddy simulation (LES) models, and mesoscale meteorological models.

Initially, based on observations, analytical models were developed in order to assess the danger of downbursts to aircrafts (Bowles and Frost, 1987; Vicroy, 1991). Such studies were, in a large part, motivated by several commercial airline

accidents (e.g., the Delta Airlines Flight 191 crash at Dallas-Fort Worth in 1985 with 137 fatalities (Fujita, 1986)) and an earlier high-profile near miss at Andrews Air Force Base in 1983 in which Air Force One carrying then-President Reagan landed just six minutes before wind speeds had reached 67 m s$^{-1}$ at 5 meters above the surface (Fujita, 1985). Developments stemming from these early analytical models proved to be useful in wind engineering studies (Holmes and Oliver, 2000; Chay et al., 2006; Chen and Letchford, 2004) and to wind energy research (Nguyen et al., 2010, 2011, 2013; Nguyen and Manuel,

2013, 2015). These relatively simple models are very efficient and inexpensive to run at high resolution making them the state of practice in the wind engineering community. However, these models fall short of representing the underlying physics accurately and, especially with regard to turbulence, they rely on synthetically generated stochastic fields associated with target power spectral density functions, which are not specifically associated with downbursts.

RANS simulations of downburst winds (Mason et al., 2009; Proctor, 1988; Mason et al., 2010a) and LES studies (Vermeire

et al., 2011a; Anabor et al., 2011; Orf et al., 2012, 2014) both solve the Navier-Stokes equations and have been used to study the dynamics of downburst winds. What separates the two methods is that RANS models estimate the entire turbulent contributions of the flow field, while LES resolves the largest turbulent eddies and parameterizes only the smaller turbulent eddies. An advantage of both approaches is that several fundamental physical features of downburst flows can be simulated. For example, RANS studies performed by Proctor (1988) produced, to the best of our knowledge, the first simulation of what is

known as the *ring vortex*—a distinctive feature commonly observed in downbursts (Fujita, 1985; Smith, 1986; Kessinger et al., 1988; Wakimoto, 2001). This feature is characterized by the rotation (roll-up) about a horizontal axis located towards the edge of the outflow and commonly associated with the strongest outflow winds. Further, the influence of various other factors such as terrain (Mason et al., 2010b), the interacting collision of two downburst outflows (Vermeire et al., 2011b), the microphysics and ambient humidity (Proctor, 1989, 1988; Srivastava, 1985, 1987) have all been explored. Full-storm simulations utilizing

advanced parameterizations (Orf et al., 2012, 2014) are now able to help us advance our knowledge of downburst dynamics and study these various influences. A major drawback of RANS, LES, and full-storm simulations is the computational expense involved in obtaining high-resolution spatio-temporal representations of downbursts. As a result, these models are rarely run at resolutions suitable for use as inflow in engineering models.

To the best of our knowledge, all previously published numerical studies of downburst winds have only considered down-

bursts in the NBL with one exception—a sensitivity study by Proctor (1989) that used a RANS model. In that study, surface-based deep inversion layers of 500 m and 1,000 m were used to mimic nocturnal stability or the outflow of another storm. It was shown that the stable layer at the surface reduced the magnitude of the outflow wind speed and the maximum downdraft; it also prevented one downburst event from even reaching the surface entirely. The height of the ring vortex that formed was also shown to be sensitive to the surface-based inversion in that this height was equivalent to (or the same as) that of the stable

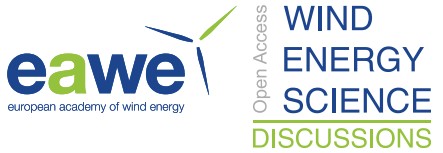

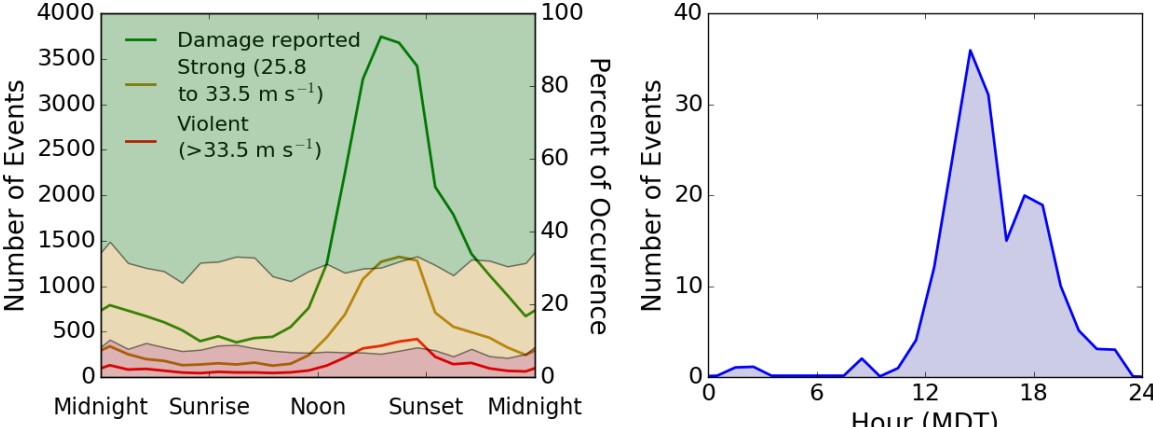

**Figure 1.** (a) The frequency of wind gusts of varying severity throughout the day (solid lines) overlaying the percentage of each wind gust severity category. Red, yellow, and green coloring correspond to violent gusts (greater than 33.5 m s$^{-1}$), strong gusts (between 25.8 and 33.5 m s$^{-1}$), and events with reported damage but no estimated wind speeds, respectively. Data have been adapted from Kelly et al. (1985) where the times have been converted to normalized solar time (NST). (b) Time of occurrence of the downbursts observed in the JAWS campaign; regenerated data from Figure 4 in Wakimoto (1985).

layer. This single study only describes part of the influence of the SBL; the expected characteristic increase in low-level wind shear, a fundamental feature of the SBL, was ignored.

When considering the three PBL regimes, it is to be expected that initial studies would only consider neutral conditions. The NBL provides the most simplistic ambient environment in that it lacks several complex processes inherent to the CBL and SBL. However, since thunderstorms and downbursts occur throughout the night and day, it is necessary to assess what effect stability might have on downburst outflows. In the present study, a pseudo-spectral LES model with a dynamic sub-grid scale (SGS) scheme is utilized to simulate downbursts in various stability regimes. The goal of this study is to analyze outflow wind fields to learn how downburst intensity and flow characteristics change with increased low-level stability. Section 2 introduces the methodology employed in this study. Results for the ET and the downburst simulations are presented in Sections 3.1 and 3.2, respectively. Finally, a summary and conclusions are provided in Section 4.

## 2   Methods

Simulating the diurnal cycle has been a challenging topic in boundary layer meteorology for some time. In convective simulations, the model domain size must be large enough in both the horizontal and vertical extent to capture the convective processes. Due to these large domains, modelers are typically constrained to using relatively coarse grid sizes to decrease the computational burden of each simulation. However, when simulating the SBL, the model grid spacing and time step must be small enough to capture developing nighttime eddies that are on the order of meters. In order to satisfactorily simulate both



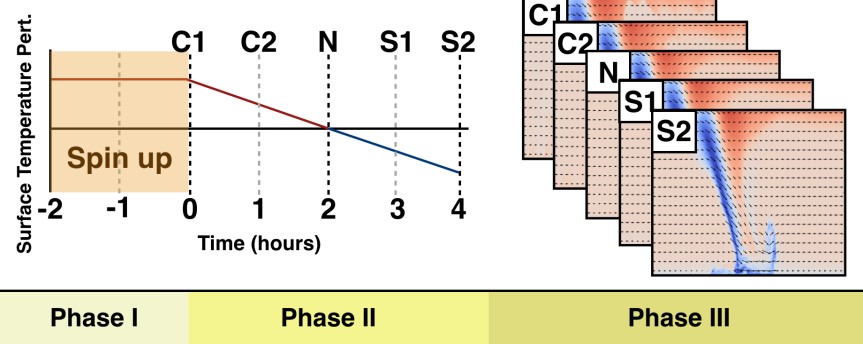

**Figure 2.** A schematic diagram depicting the process for simulating downbursts throughout the evening transition period. The first phase consists of model spin up to generate the CBL. The second phase simulates the ET while outputting a snapshot every hour to be used as the initial and boundary conditions in the downburst simulations performed in Phase III. These snapshots are denoted as C1 for the moderately convective case, C2 for the weakly convective case, N for the neutral case, S1 for the weakly stable case, and S2 for the moderately stable case.

regimes, a large model domain with small grid spacing is necessary; this results in the need for a large amount of computational grids. With advances in computational power, we have began to be able to run these types of simulations with relative ease.

Basu et al. (2008) used a dynamic SGS scheme, called the locally-averaged scale-dependent dynamic (LASDD) model (Basu and Porté-Agel, 2006), to simulate a full diurnal cycle. The results showed good agreement with observations and, additionally,
showed little dependence on model grid spacing. In the LASDD SGS model, the SGS coefficients are computed dynamically based on the local dynamics of the flow, whereas in typical SGS schemes the SGS coefficients are specified in an *ad hoc* manner.

The LES model used in Basu et al. (2008) is utilized here to simulate the idealized downbursts. The model forcing is imposed by specifying the geostrophic wind and the surface temperature. The model domain spans 10 km in both lateral directions and
3 km in the vertical. The grid spacing is approximately 28 m in both the horizontal and vertical directions. The model's lateral boundaries are periodic in both the x- and y-direction, and a damping layer is employed on the top 1 km of the domain in order to eliminate reflections off of the model top. A roughness length of 0.1 m is utilized corresponding closely to the average roughness length over North America (Stull, 1988). The simulations of downbursts during the evening transition is conducted in three phases (Figure 2).

**2.1   Phase I: Spin-up Simulation**

The model is initialized with a neutral potential temperature profile up to the initial boundary layer height of 1.5 km, and a constant geostrophic wind of 8 m s$^{-1}$ in the positive-$x$ direction (i.e., $U_g = 8.0$ m s$^{-1}$; $V_g = 0.0$ m s$^{-1}$). A prescribed surface temperature is used as the lower boundary condition to drive the flow. In this case, the initial temperature in the NBL is 300 K and the surface temperature is prescribed to be 303 K. For the first two hours of the simulation, the constant surface temperature




is used to develop the convective boundary layer. The model time step, $\Delta_t$, is set to 0.5 seconds. These two hours are regarded as the period of model spin-up and data from this phase are not used in subsequent analysis.

## 2.2 Phase II: ET Simulation

In the second phase, the ET is simulated for four hours where the surface temperature is decreased by 1.135 K hr$^{-1}$ such

that after two hours the boundary layer will reach a neutral state and the SBL will be developed in the subsequent two hours. The selected surface temperature cooling rate was found by trial and error in order to produce a neutral state after two hours of cooling. In this phase, the same $\Delta_t$ of 0.5 seconds is used as in Phase I and, after every hour of simulation, the full 3-dimensional velocity and temperature fields are output to be used as initial and boundary conditions for Phase III simulations. These snapshots are denoted as C1 for the moderately convective case, C2 for the weakly convective case, N for the neutral

case, S1 for the weakly stable case, and S2 for the moderately stable case; they are output at hours 0, 1, 2, 3, and 4, respectively.

## 2.3 Phase III: Downburst Simulations

The downburst simulations are initialized from the C1, C2, N, S1, and S2 snapshots of the ET simulation. In these simulations, $\Delta_t$ is reduced to 0.1 seconds and each simulation is run for 15 minutes. Due to the sensitivity of LES runs to initial conditions, an ensemble of simulations is generated for each stability case (C1, C2, N, S1, and S2). Each case is run four times where,

taking advantage of the periodic boundaries in the $x$- and $y$-directions, the initial conditions generated from the ET simulation are shifted as follows: (1) in the $x$-direction by +5 km (denoted by 5x_0y), (2) in the $y$-direction by +5 km (denoted by 0x_5y), and (3) in both the $x$- and $y$-directions by +5 km (denoted by 5x_5y). Including the original initial conditions, this strategy results in four sets of initial conditions for each case. Shifting the initial conditions in this manner effectively initializes the downburst at different locations within the same ABL regime.

## 2.4 Downburst forcing

A cooling source near the top of the model originally proposed by Anderson et al. (1992) is used to produce negatively buoyant air to simulate the effects of latent cooling due to the melting, evaporation, and sublimation of hydrometeors as they descend towards the surface. These processes are largely responsible for generating the negative buoyancy that drives the downburst (Wakimoto, 2001) and this approach has become the state of the practice in LES simulations of downburst winds

(see, for example, Orf et al. (1996); Anabor et al. (2011) or Oreskovic (2016)). The source ($Q$) equations are as follows:

$$Q(x,y,z;t) = \begin{cases} C_{\max} \times g(t)\cos^2(\pi R), & R \leq \frac{1}{2} \\ 0, & R > \frac{1}{2} \end{cases} \tag{1}$$



where $C_{\max}$ is the peak intensity of the cooling function and is set to -0.08 K s$^{-1}$ as was done in Mason et al. (2009) and Anabor et al. (2011) to produce an intense downburst. The function, $g(t)$, defines the variation of downburst intensity with time as follows:

$$g(t) = \begin{cases} \cos^2\left[\pi\left(\frac{t-120}{2\tau}\right)\right], & 0 < t \leq 120 \\ 1, & 120 < t \leq 240 \\ \cos^2\left[\pi\left(\frac{t-240}{2\tau}\right)\right], & 240 < t \leq 360 \end{cases} \tag{2}$$

5   where $\tau$ is set to 120 seconds. Equation 2 differs from previous studies in that the cooling source is held constant for 2 minutes, as opposed to 10 minutes. This is done in order to analyze the decay of the downburst events before they reach the simulation boundary as well as to limit the source from generating multiple strong downburst events. Finally, $R$ represents the normalized distance from the center of the cooling function (Vermeire et al., 2011a; Anabor et al., 2011):

$$R = \sqrt{\left(\frac{x-x_f}{M_x}\right)^2 + \left(\frac{y-y_f}{M_y}\right)^2 + \left(\frac{z-z_f}{M_z}\right)^2} \tag{3}$$

10   Here, the coordinates of the center of the downburst, $(x_f, y_f, z_f)$, are set to (2.5 km, 5.0 km, 1.9 km). The $x$-position is chosen so as to allow the downburst to reach the surface near the center of the domain and extend the amount of time before the downburst reaches the model boundary. The vertical location of the cooling center is chosen such that it is concentrated just below the entrainment zone overlying the mixed layer. The horizontal and vertical extent of the cooling source, defined by $M_x$, $M_y$, and $M_z$, are set to 2.0 km, 2.0 km, and 1.5 km, respectively, such that the vertical extent of the cooling source does 15   not reach the surface. This type of cooling source essentially mimics latent cooling from a *dry* microburst in which most, if not all, of the hydrometeors melt, evaporate, and/or sublimate before reaching the ground.

## 3   Results

A brief analysis of the evening transition simulation (which provides the initial conditions for the downburst simulations) is presented in Section 3.1, followed by a detailed analysis of the downburst cases in Section 3.2. The analysis performed herein 20   is from the lowest two kilometers of the simulation domain in order to limit any influence from the damping layer.

### 3.1   ET simulation

The evening transition simulation produces hourly restart files to be used as the initial conditions for each of the downburst simulations. The potential temperature and wind velocity profiles for each case are shown in Figure 3. As can be seen, both the C1 and C2 temperature profiles (red and yellow lines, respectively) are unconditionally unstable throughout the boundary 25   layer with the instability reducing to zero by case N (grey line). Even though the surface temperature is decreasing between the C1 and C2 cases, the prescribed temperature is still larger than that of the mixed layer. The warmer surface combined with





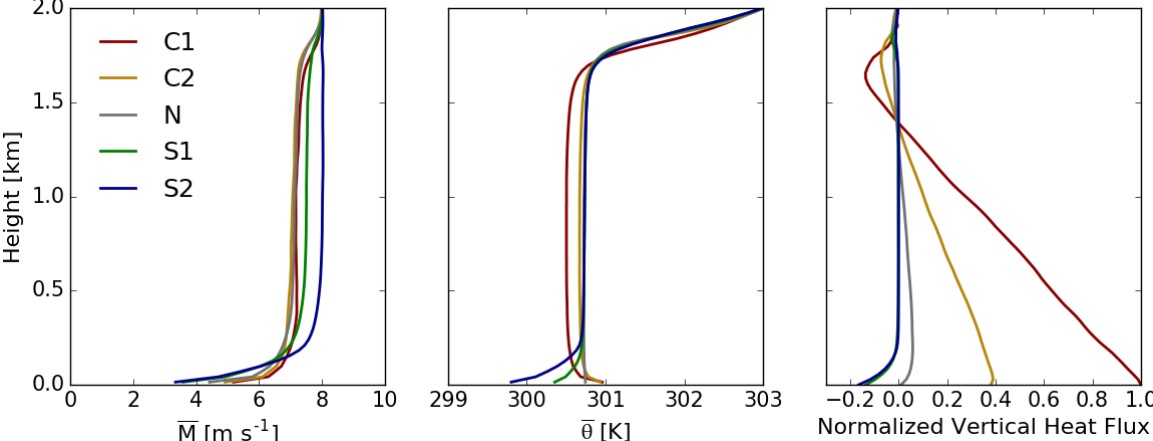

**Figure 3.** Vertical profiles of average wind speed (left), temperature (center), and normalized vertical sensible heat flux (right) are shown for the C1, C2, N, S1, and S2 cases colored red, yellow, grey, green, and blue, respectively. The normalized vertical sensible heat flux profiles are normalized with respect to the average surface sensible heat flux of the C1 simulation.

the warming from entrainment (mixing of the warm air above the boundary layer into the mixed layer) results in a continual warming in the mixed layer between C1 and C2. As the simulation progresses, a surface based stable layer develops by S1 and deepens to S2 (green and blue lines, respectively). The mixed layer temperature between the C2 and N cases continues to subtly warm. In previous ET studies, it has been shown that entrainment can persist due to residual convection after the surface

heating has stopped (Nieuwstadt and Brost, 1986; Sorbjan, 1997). During the generation of the stable layer, the average wind speed throughout the residual layer (remnant of the mixed layer) increases along with the deepening of shear below roughly 250 m. This low-level stable layer is estimated to be as deep as 200 m.

The vertical sensible heat flux profiles are normalized relative to the surface sensible heat flux of the C1 case. It can be seen here that over time the amount of vertical sensible heat flux near the surface and within the mixed layer decreases to zero by

the N case and becoming negative after. The decrease of vertical sensible heat flux throughout the boundary layer occurs from the bottom up as is consistent with the literature (Nieuwstadt and Brost, 1986). The ET is defined as the period in which the boundary layer becomes approximately neutral near the surface and achieves a vertical sensible heat flux of zero at the lowest model level. The negative portion of vertical sensible heat flux above roughly 1.5 km is caused by entrainment. The height at which the vertical sensible heat flux is most negative is often used to define the daytime boundary layer height. Entrainment

acts to increase the boundary layer height most notably between C1 and C2.

## 3.2   Downburst Simulations

The downburst simulations are each run for 15 minutes. After about 13 minutes, the outflow reaches the domain boundary in each simulation. For this reason, only data from the vertical plane at x= 6.0 km are analyzed for the full 15 minutes as the boundaries do not influence these results. Between the different downburst simulations, only the initial and boundary conditions





are changed. Surface temperature is decreased, as within the ET simulation, for the full downburst duration. For generalization purposes, the ensemble mean for several of the analysis fields is presented here. However, in order to gain insight into the individual downbursts, a characteristic simulation is chosen from each case for certain analyses. Downburst simulation results are presented by referring to time in minutes after the case initiation, denoted by $\text{Time}_{\text{Burst}}$. The hours into the ET simulation

at which the downburst cases are initiated can be seen in the schematic presented in Figure 2.

### 3.2.1 Velocity Fields

In each of the downburst simulations, a diverging outflow pattern in the horizontal wind fields is generated as shown in Figure 4. The initial (top row) and mature-to-dissipating (bottom row) stages of the downbursts are shown for the C1–5x_5y (left column), N–5x_0y (center column), and S2–0x_5y (right column) cases. These horizontal cross-sections are taken at

around 100 m above ground level; a typical value for the hub height of a commercial wind turbine. The initial stages of the downbursts show large swaths of areas with horizontal winds greater than 20 m s$^{-1}$ (denoted by black arrows), mostly confined to regions downwind of the downburst (i.e., to the right of the downburst center). At this stage, the C1 and N cases appear to generate generally stronger horizontal winds than the stable, S2, case. However, below 100 m, the wind speeds for *all* cases increase (not shown), suggesting that the wind speeds in the S2 case are simply confined to lower heights than the other

cases. As the outflow evolves, by $\text{Time}_{\text{Burst}} = 10.25$ minutes, the S2 case has fully developed a large, strong horizontal outflow region while the convective case has begun to dissipate. It can be seen that the ambient winds in the C1 case have much more variability than the N and S2 cases, which effectively deforms and disorganizes the outflow.

The large regions of high winds extending generally in the North-South direction in Figure 4 are organized by the ring vortex. This feature can be seen near the vertical dashed line in Figure 5 for each of the presented cases. The ring vortex increases

the wind speed beneath it; the center of this ring vortex occurs at increasingly higher elevations as stability increases. The depth of the head of the outflow is also seen to increase with increasing stability due to the interactions of the ambient winds with the outflow—a similar result to what has been reported based on RKW theory (see Rotunno et al. (1988) for details). This theory shows how low-level shear can interact with outflow to generate stronger, more vertically upright updrafts from the counteracting vortices from the outflow and the ambient environment. Interestingly, a similar result is also reported in the

simulations by Proctor (1989) where, by only introducing a surface-based temperature inversion, the depth of the outflow and the height of the ring vortex increase to equal the depth of the stable layer. In the present environment, both an increase in low-level shear and a surface-based temperature inversion are considered and a qualitatively similar trend is observed.

The vertical wind shear generated from the development of the ring vortex can easily be seen in the vertical profiles of horizontal (black) and vertical (blue) winds in Figure 5 (right panels). These profiles are taken at the center of the domain

in the y-direction and at $x = 6.0$ km right after the passage of the center of the ring vortex. A local minimum in the vertical velocity profile can be seen at around 250 m, 400 m, and 550 m for the C1–5x_5y (top row), N–5x_0y (middle row), and S2–0x_5y (bottom row) cases, respectively. This local minimum is due to the downward motions associated with the upwind side of the ring vortex and occurs at the height of the center of the ring vortex. As previously mentioned, the increases in stability cause the height of the center of the ring vortex to increase. While the N–5x_0y and S2–0x_5y cases show a stronger

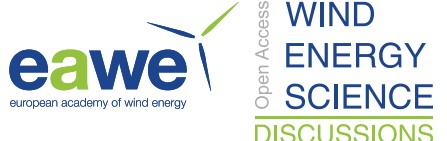

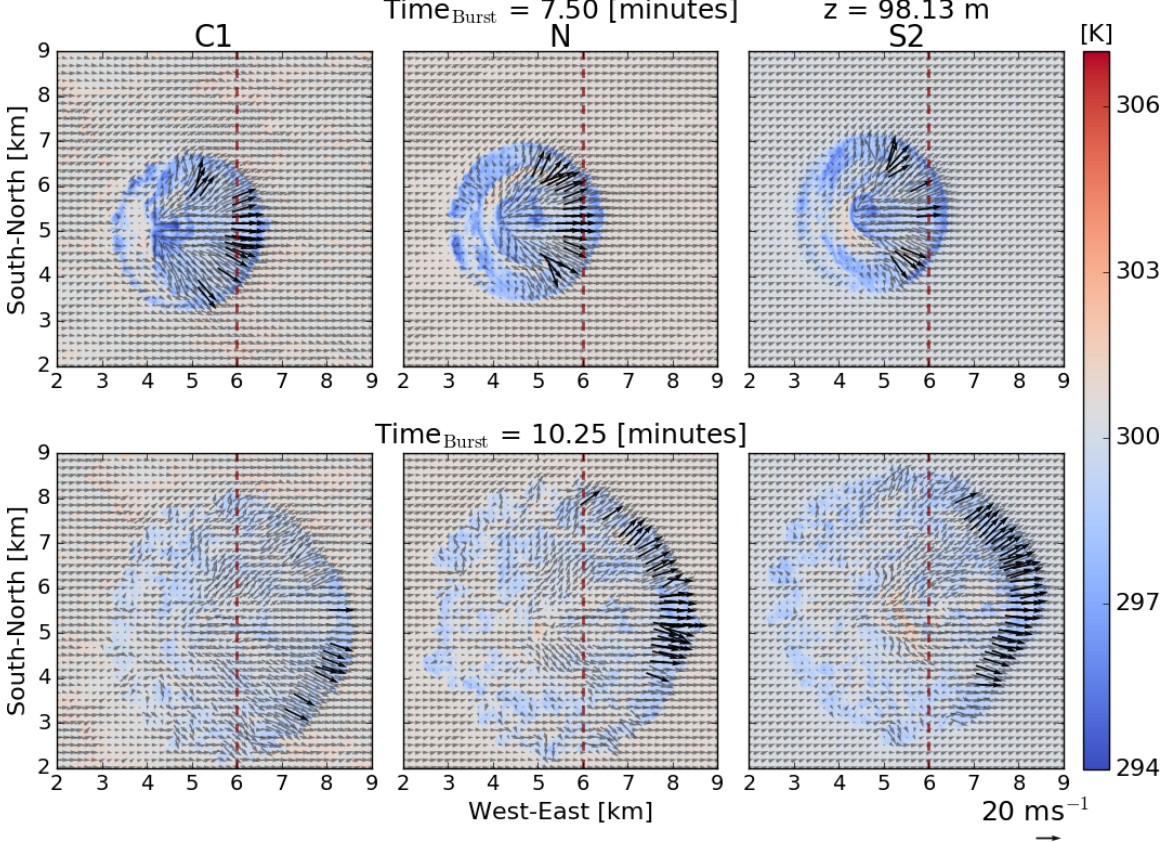

**Figure 4.** Horizontal cross sections at 98.13 m for the C1–5x_5y (left column), N–5x_0y (center column), and S2–0x_5y (right column) cases at $\text{Time}_{\text{Burst}}$ = 7.5 minutes (top row) and 10.25 minutes (bottom row). Filled contours show the temperature [K] with arrows representing the horizontal wind field at this level. The locations with wind speeds above 20 m s$^{-1}$ are shaded in black, while locations with wind speeds below this threshold are shaded in grey. The North-South oriented line denotes the location at which a vertical slice is recorded at every time step used to generate the time series in Figure 6.

downdraft than the C1–5x_5y case, this apparent strengthening of the downdraft is most likely due to the decreased distance from the center of the ring vortex in the N–5x_0y and S2–0x_5y cases. The vertical structure in the horizontal winds in case C1–5x_5y shows an increasing horizontal wind from the height of the ring vortex down to the surface. Cases N–5x_0y and S2–0x_5y, however, show a vertical structure below around 250 m that is quite complex. The wind speeds decrease from the

5   lowest level to around 150 m, then sharply increase again before quickly decreasing to ambient wind speeds at 500 m. The strongest horizontal wind speeds occur at the lowest model level in these cases, however, the secondary peak above this appears to be associated with the temperature gradient generated by the warm air entraining into the ring vortex. It should be noted that these complex wind patterns are occurring at heights that would impact typical commercial wind turbines.





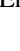

**Figure 5.** Vertical cross sections along the East-West axis at the center of the domain for the C1–5x_5y (top row), N–5x_0y (middle row), and S2–0x_5y (bottom row) cases at Time$_{Burst}$ = 7.5 minutes. Filled contours show temperature [K] with arrows representing the u-w wind field. In the right column are the horizontal velocity and vertical velocity profiles (black and blue, respectively) for each case at the dashed line in the vertical cross section (of the left column figures).





To get an idea of the temporal structure at a single point during the passage of the downburst, Figure 6 shows the magnitude of velocity (M; black line) and the individual components of the wind field (u, v, and w) in red, green, and blue, respectively, at $y_{max}$. Here, $y_{max}$, is defined as the location along the vertical cross sections at which the maximum horizontal wind speed occurs at 98.13 m above the ground over the full 15 minute duration. This location will change from case to case as

the maximum wind speeds along this cross-section do not always happen at the same location. The time series are shown at 98.13 m for direct comparison with Figure 4 for each of the simulations in cases C1 (top row), N (middle row), and S2 (bottom row). Each of the ensemble members are shown to highlight the extent of the deviation between runs. By studying the vertical velocity time series (blue line) for each case, the time between peak-positive and peak-negative vertical wind speeds (indicative of the ring vortex passage) is seen to be as short as 20 seconds to as long as about 1 minute. Generally, this time between these

positive and negative peak values increases as stability increases. This is due to the associated increased spatial extent of the ring vortex or due to a decrease in the speed at which it propagates (Proctor (1989) found that a low-level temperature inversion decreased the propagation speed of the outflow). Each case also shows the maximum wind speed occurring in between the maximum updraft and maximum downdraft, consistent with the passage of the ring vortex.

In the C1 simulations, the vertical velocity and lateral velocity components are quite low relative to the other simulations.

This is because $y_{max}$ is located closer to the center of the domain and because a weaker ring vortex forms in these convective simulations. In the neutral simulations, the $y_{max}$ locations are split between north of center and south of center (indicated by the positive and negative peaks, respectively, in the v-component wind speeds), hinting at a lack in preferential location of occurrence of the maximum wind speed. On the other hand, in the S2 simulations, all of the $y_{max}$ locations are north of center where the outflow propagation direction is *normal* to the direction of the low-level wind shear vector. A similar result

was found in a numerical case study by Proctor (1994) of a dry, pulsating microburst event. In the current model, the low-level wind shear is generated by the Coriolis effect on the low-level stable layer causing the winds to veer with height; a characteristic feature of the SBL (Stull, 1988). This is consistent with the added ambient shear in the stable cases that bolsters the ring vortex and, thus, helps to increase horizontal wind speeds at this location. Here, we also see that the peak horizontal wind speed reaches the vertical cross-section at x = 6.0 km earlier in the C1 simulations and later in the S2 simulations. Although there

are differences in the propagation speed from case to case, the differences in timing here are due to the circular nature of the downburst resulting in the ramp reaching locations along the vertical slice at different times. The location of $y_{max}$ changes in each case with the S2 cases showing a preference to locations along the slice further from the center of the domain which result in the wind ramp occurring at a later time than the C1 and N cases. Lastly, the additional stability in the S2 cases allows the wind field to recover to pre-outflow values more rapidly for the stable and neutral cases; this recovery is thus a function of

the stability as well as the location of $y_{max}$.

The ensemble mean of the domain-maximum horizontal wind speeds at each height with time is presented using filled contours in Figure 7. Open contours of the ensemble mean of the maximum downdraft (black) and maximum updraft (dotted magenta) are also included in this figure in order to deduce the state of the downburst. Several generalizations can be made about the structure of the maximum wind speeds for each of the simulations. First, the initial signal of the downburst comes

from a rapid increase in intensity of the downdraft. Then, horizontal wind speeds near the surface begin to increase; this is



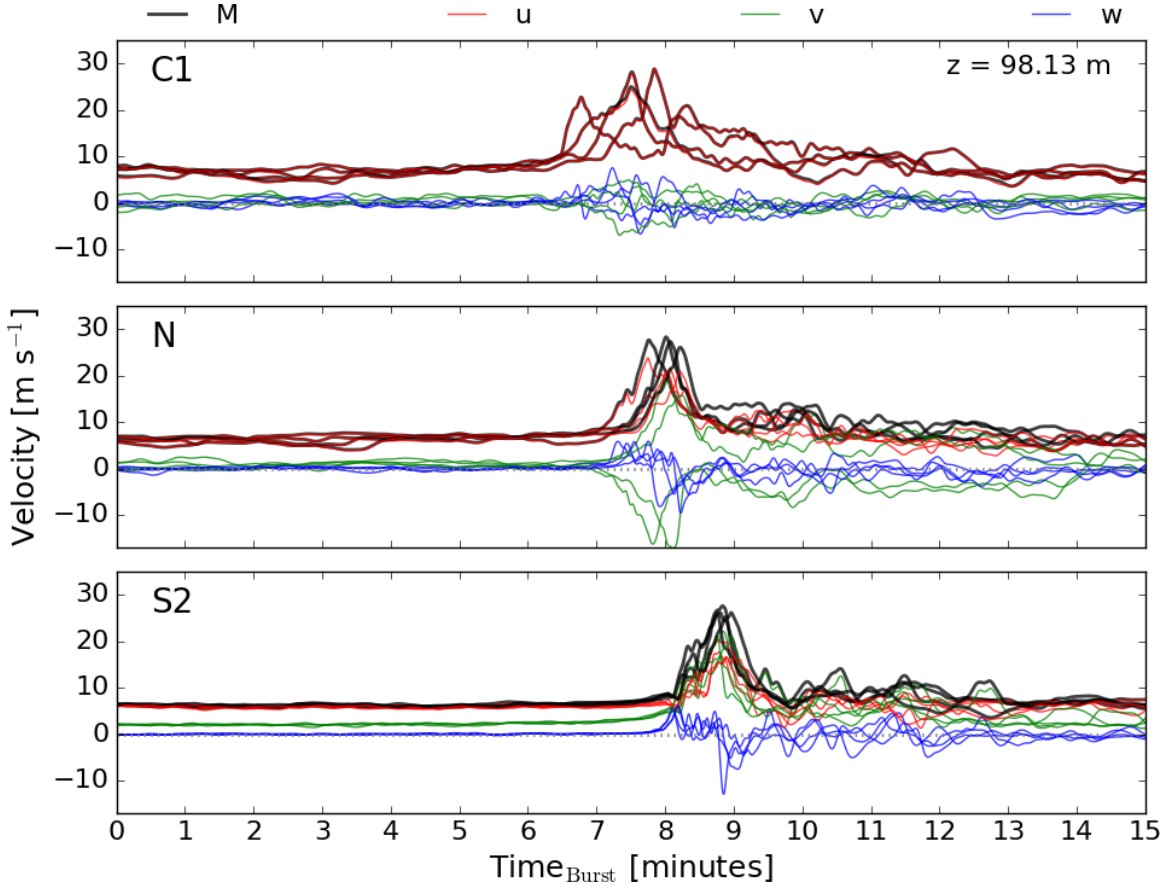

**Figure 6.** Time series of the velocity components at the location (x = 6 km, y = $y_{max}$, z = 98.13 m) for the C1 (top row), N (middle row), and S2 (bottom row) cases at the 6.0 km cross-section (red dashed line in Figure 4).

followed by a strengthening of the updraft wind speeds between around 100 m to 300 m. The peak wind speed is generally reached just after the updraft band develops with the exception of the C1 case where the two appear to occur simultaneously in the ensemble averaged output. Finally, as the updraft band begins to vertically expand, the horizontal outflow decreases. This happens at different times and rates for each case. The collocation of the updraft and downdraft bands is due to the ring vortex.

5  When these bands expand and taper off, it is indicative of a ring vortex that has become disorganized and begun to dissipate.

Inspecting C1, it is clear that the strength of the downdraft is weaker than the other simulations. This is due to the interaction of convective thermals and increased turbulent motions as the downburst descends through the column (see the right panel in Figure 3). The impact of this interaction can be seen in Figure 5 where the downdraft (between roughly 3.0 km and 4.5 km in the West-East direction) is noticeably more disorganized in the C1 case but much more coherent in the N and S2 cases. By

10  $\text{Time}_{\text{Burst}} = 11$ minutes in Figure 7, the downdrafts are no longer below -10 m s$^{-1}$ and the updrafts begin to ascend farther from the surface. This shows the demise of the ring vortex and coincides with the rapid weakening of the near-surface winds.



As the stability increases, it can be seen that the maximum downdraft increases and that it also develops at a faster rate. Further, the initial depth of the strong horizontal winds produced from the outflow is contained closer to the surface with increasing stability, indicating an increase in outflow-generated shear across the low levels of the atmosphere. Decay of the strong updrafts and downdrafts occurs at later times as stability increases with case S2 showing the first signs of decay at
$\text{Time}_{\text{Burst}} = 12$ minutes.

Several downburst modeling studies have attempted to find a relationship between the maximum outflow wind speed ($U_{\text{storm}}$) to maximum downdraft ($w_{\text{min}}$) by calculating the ratio of $U_{\text{storm}}$ to $w_{\text{min}}$ (Proctor, 1989; Mason et al., 2009; Proctor, 1988; Anabor et al., 2011). Proctor (1989) showed that it was difficult to deduce any universal relationship between $U_{\text{storm}}$ and $w_{\text{min}}$ in the brief study on the low-level stability. However, in his model, microphysics were being utilized and, thus, many
more variables came into play in this scenario which makes deducing any relationships more complex. In this study, we attempt to determine relationships between this ratio and stability. Due to the peak wind speeds occurring so close to the surface, the vertical component is almost negligible, thus, in our computation of $U_{\text{storm}}$ we do not include the vertical velocity component (Mason et al., 2009).

Figure 8 shows plots of the $U_{\text{storm}}$, $w_{\text{min}}$, and the ratio of $U_{\text{storm}}$ to $w_{\text{min}}$ in the top, middle, and bottom rows, respectively,
for the different simulations: C1 (red), C2 (yellow), N (grey), S1 (green), and S2 (blue). These quantities are plotted against the bulk Richardson number (Stull, 1988) at the lowest model level and averaged over the first three minutes of the simulation to deduce their relationships with stability. In studying the maximum outflow wind speed, a subtle and almost insignificant increase can be seen with increasing stability. The spread of the values tends to decrease as stability increases as well, owing to the heterogeneity of the convective simulations transitioning to the more homogeneous, stable regime. When considering only
the ensemble mean values, as shown in Table 1, an increase with stability can be generally seen, although it is not an entirely monotonic increase. On the other hand, $w_{\text{min}}$ is clearly strengthened with increasing stability as can be seen in Figure 8 and Table 1. As previously mentioned, this is most likely attributed to the decrease in turbulence which would act to impede the downburst as it descends. In tracking the height at which the maximum downdraft occurs for each ensemble member, Table 1 shows a somewhat unclear relationship between the ensemble mean maximum downdraft and ensemble mean of the heights at
which this downdraft occurs. Here, we can see that the higher downdraft wind speeds are occurring at lower heights. Intuitively, one might expect the stable surface layer to impede the progress of the downburst at lower heights and force the peaks to occur at higher levels. The opposite is seen in the cases herein, likely because the stable layer developed in the ET simulation is too shallow to impede the downdraft.

Finally, comparing the ratio of $U_{\text{storm}}$ to the magnitude of $w_{\text{min}}$ in Figure 8, we notice what appears to be a converging pattern
to a value of roughly 1.3 as stability increases. The spread of values decreases significantly from the C1 cases to the two stable cases. Overall, the range of values of this ratio is in line with what other downburst simulations have produced ( Proctor (1988) and  Proctor (1989) showed values ranging from 0.8 to 2.4, although this was while varying microphysics; simulations by Mason et al. (2009) produced a value of 1.58 in neutral conditions; and  Anabor et al. (2011) produced a value of 1.34 in neutral conditions). It is not quite clear how universal this ratio is, as the maximum downdraft is clearly shown to depend on



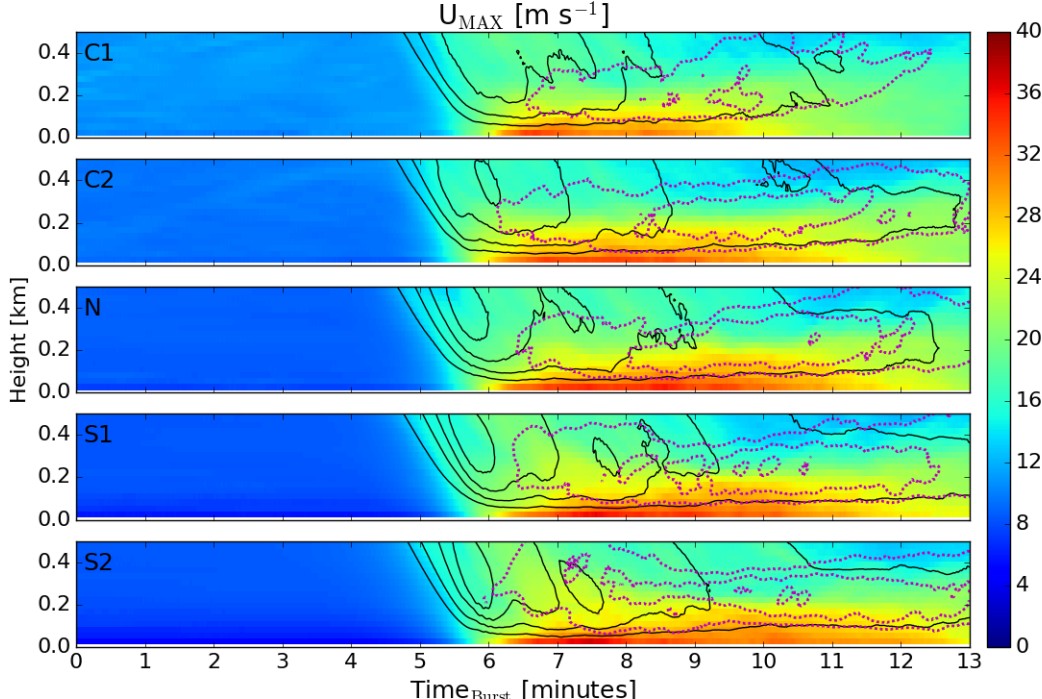

**Figure 7.** A time-height plot in the lowest 500 m of the ensemble average of maximum horizontal velocity for the C1, C2, N, S1, and S2 cases going from top to bottom. Contoured (in black) are the ensemble averages of minimum vertical velocity starting from -10 m s$^{-1}$ and drawn at -5 m s$^{-1}$ intervals and (in dotted magenta) the ensemble averages of maximum vertical velocity starting from positive 10 m s$^{-1}$ and drawn at 2.5 m s$^{-1}$ intervals.

the ambient environment and the maximum outflow will depend on the environment as well as on variables such as surface roughness.

As downbursts develop and mature, the maximum wind speeds move with the ring vortex farther away from the downburst center. To show this, the downburst center is predicted as the location of maximum divergence over a roughly 1 km diameter area in both the x- and y-directions and is calculated every 15 seconds. This process occasionally results in large jumps in the predicted downburst center location between model time steps so in order to ameliorate the prediction, a linear regression is performed for the predicted centers between Time$_{\text{Burst}}$ = 6 and 10.5 minutes and extrapolated to all times. Next, the maximum wind speeds at the lowest model level are recorded for each simulation in bins of 100 m expanding radially from the downburst center. In this way, we are able to track how the downburst expands and how the velocity field changes with radial distance from the center and with time as shown in Figure 9. Here, only wind speeds to the right of the downburst center (downstream) are considered for the C1 (top row), N (middle row), and S2 (bottom row) cases with each ensemble member (columns 1 through 4) and the ensemble average (far right column) shown. It is clear that in each of the C1 simulations, the outflow begins





**Table 1.** Ensemble-averaged maximum event horizontal wind speed, ensemble-averaged maximum event downdraft, and the ratio of those two for each case, along with the ensemble average of the time from first sign of divergence at the surface ($\nabla \cdot U_{\text{sfc}} = 1.5 \nabla \cdot U_{\text{sfc}_0}$) to the time of maximum surface wind speed.

| Case | $U_{\text{storm}}$ [m s$^{-1}$] | $w_{\text{min}}$ [m s$^{-1}$] | $|U_{\text{storm}}/w_{\text{min}}|$ | Height of $w_{\text{min}}$ [km] | $t_{\text{ramp}}$ [minutes] |
|------|------|------|------|------|------|
| C1 | 35.86 | -25.98 | 1.38 | 0.371 | 1.54 |
| C2 | 35.30 | -26.82 | 1.32 | 0.343 | 2.44 |
| N  | 37.03 | -27.71 | 1.34 | 0.364 | 3.58 |
| S1 | 37.02 | -28.22 | 1.31 | 0.364 | 3.07 |
| S2 | 38.31 | -29.89 | 1.28 | 0.315 | 2.91 |

to weaken earlier on in the simulations than for the neutral and S2 cases. C1-0x_5y, for example, appears to deteriorate just after reaching the surface. The length of time over which the outflow persists appears to be related to the ambient stability. The ensemble mean for the S2 case shows what appears to be a rather constant-intensity wind speed of around 30 m s$^{-1}$ that persists past 12 minutes while the other cases show considerable weakening by this time. The propagation speed of the downbursts

also appears to change with stability. In the S2 case, the peak wind speeds initially move away from the downburst center at a much quicker rate than for the other cases. However, after a minute or so, this propagation away from the center begins to slow to a pace similar to that of the C1 ensemble mean. Proctor (1989), being the only study in which a stable regime has been tested that we are aware of, showed that a low-level temperature inversion acts to weaken the outflow wind speed, slow the propagation rate, and decrease the peak downdraft. In the simulations herein, the opposite occurs (see Table 1). This could

be due to several factors: first, the depth of the stable layer is much shallower in these simulations ( Proctor (1989) considered an artificial temperature inversion through a depth of 500 m and 1,000 m); second, the inclusion of low-level shear in these simulations appears to play a significant role in maintaining the ring vortex and, thus, generating a more persistent outflow; and third, Proctor (1989) utilizes a RANS model, whereas an LES model is utilized herein. It is also speculated that with a shallow surface-based stable layer such as in this current study, it may be possible for the increased negative buoyancy generated from

the lifting of the colder air at the surface to further strengthen the outflow and ring vortex in the stable cases.

Upon studying the ensemble average for each case, it is clear that the C1 case increases to its peak velocity much quicker than the N and S2 cases. For each case, the time from which divergence at the lowest model level first increases past one and a half times the average, environmental surface divergence, $\nabla \cdot U_{\text{sfc}} = 1.5 \nabla \cdot U_{\text{sfc}_0}$, to the time at which the maximum wind speed occurs, $t_{\text{max}}$, is calculated and presented in Table 1 under $t_{\text{ramp}}$. The variable, $t_{\text{ramp}}$, shows that the convective cases reach peak

velocity quicker than in the neutral and stable cases, perhaps because the turbulence disorganizes the outflow more rapidly. However, it is interesting to note that the neutral case takes the longest time to reach peak velocity. On average, the S2 cases reach their peak velocity in less time than the S1 cases. We speculate that this is due to the ring vortex developing quicker with increased shear or potentially due to the additional negative buoyancy ingested from the surface-based stable layer.

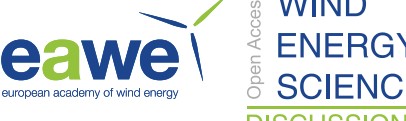



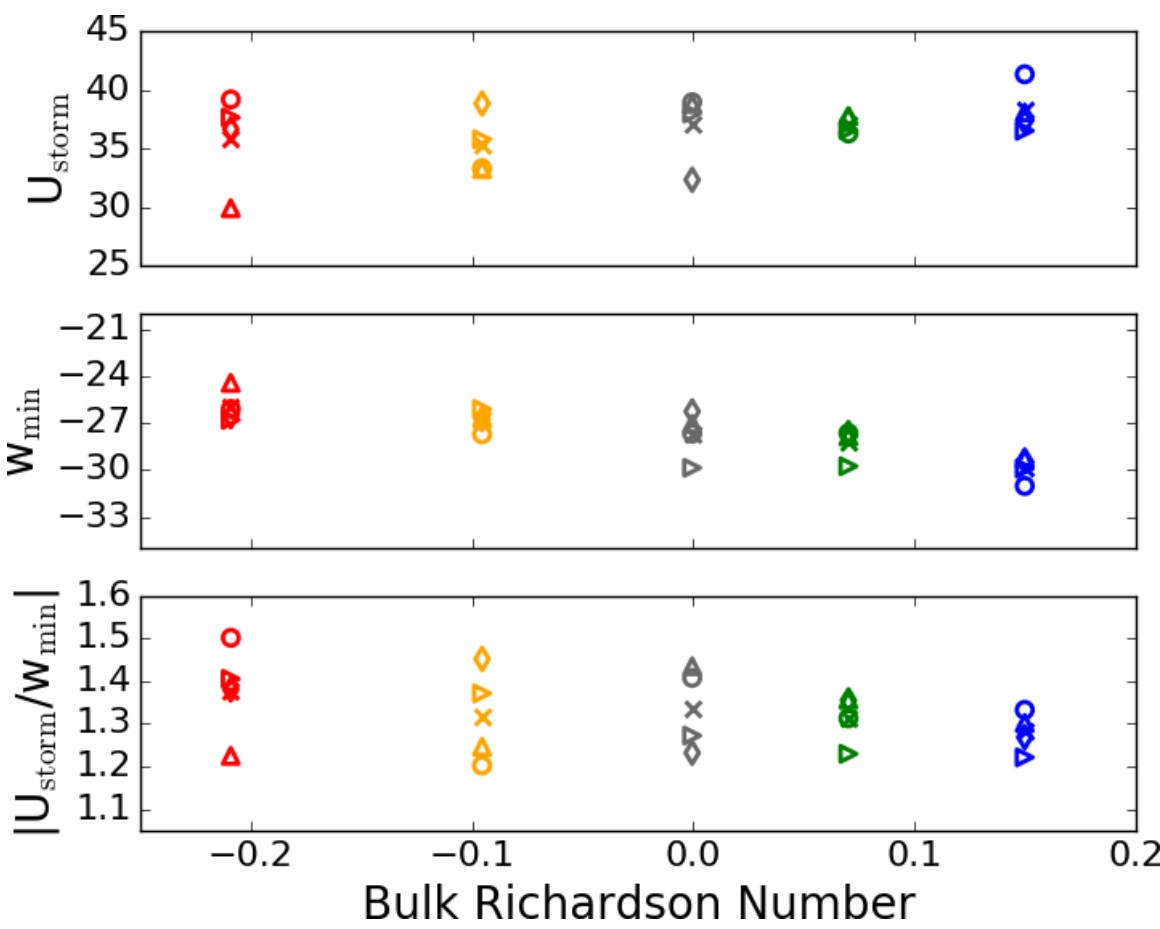

**Figure 8.** Variations of $U_{storm}$ (top), $w_{min}$ (middle), and the ratio between the two (bottom) for each case—C1 (red), C2 (yellow), N (grey), S1 (green), and S2 (blue)—with the average bulk Richardson number at the first model level over the first three minutes of simulation. Circles represent the 0x_0y cases, the right-pointing triangles represent the 5x_0y cases, the upward-pointing triangles represent the 0x_5y cases, and the diamonds represent the 5x_5y cases. The ensemble mean is denoted by an "x" for each case.





**Figure 9.** Variation with time of the horizontal wind speed [m s$^{-1}$] at the lowest model level at binned radii to the right of the downburst center for cases C1 (top row), N (middle row), and S2 (bottom row). Each case is presented in the first four columns while the ensemble mean is shown in the right column.



### 3.2.2 Ensemble-averaged variances

Throughout each simulation, domain-averaged profiles of several resolved variances, $\sigma^2_{u_i}(t,z)$, $\sigma^2_{v_i}(t,z)$, and $\sigma^2_{w_i}(t,z)$, are calculated and output every second for each ensemble member, $i$, in order to analyze how the downburst wind fields modify the environment. Note that with increased resolution, it is expected that the resolved variances will increase in the S1 and S2 cases.

The resolution achieved in this model setup is adequate for convective and neutral simulations. Next, the ensemble averages, $\langle\sigma^2_{u_{\text{Ens}}}\rangle(t,z)$, $\langle\sigma^2_{v_{\text{Ens}}}\rangle(t,z)$, and $\langle\sigma^2_{w_{\text{Ens}}}\rangle(t,z)$, are computed for each case to get a generalized result. The profiles of maximum variance for the background (dotted) and the downburst (solid) are reported in Figure 10. In terms of the environment for the maximum u- and v-component ensemble-averaged variances, the magnitude of these variances is seen to decrease with increasing stability. However, with the passage of the downburst, the peak variances occur near the surface and increase in

magnitude as stability increases (note that due to the profiles being first domain-averaged, the variance generated by the downburst and the ambient environment are both included in these values). Both the u- and v-component variances return very close to their ambient values by around 400 m and only truly begin to deviate and increase below 200 m. The right panel shows the maximum of the ensemble-averaged vertical velocity variance where a much more interesting feature is seen to develop. In the most convective case (C1), the maximum vertical velocity variance appears to increase while generally retaining the

shape from the ambient profile. However, as the environment becomes more stable, a nose-like profile begins to form. First, noticeable in C2, the nose increases in value (but not in height) to case N. As the stable layer develops, the peak begins to increase in height (but not much in value) from the N to the S2 case. Above this nose, the maximum begins to decrease as the environment becomes more stable. As opposed to the variances from the other two components of the wind field, the vertical velocity variance is the lowest at the surface, as one would expect, but grows significantly even with the influence of

the ambient environment in the domain-averaged profile and reaches its peak value between around 160 m and 225 m. The combination of these variances being so large and their being so highly sheared within the lowest 200 m of the atmosphere further emphasizes the dangers of downburst winds to structures such as wind turbines.

In order to determine the cause of the nose-like structure in the maximum of the ensemble averaged vertical velocity variance, Figure 11 shows the variation with time of the ensemble mean of the vertical velocity variance profiles for each case. As in

Figure 7, open contours of negative (positive) vertical velocity in black (dotted magenta) are included for reference. At the time of the strongest downdraft, the ensemble-averaged vertical velocity variance is large for the convective cases but much weaker for the stable cases. This is due partly to the increased ambient vertical velocity variance in the convective cases as can be seen in minutes 1–5 of the simulations, as well as to the increased mixing from the cool downburst and warm thermals. As the outflow develops in each case, the vertical velocity variance shows a clear area of maximum variance coincident with the areas

of large positive and negative vertical velocity. It is at this height that the nose-like feature exists and is due to the formation of the ring vortex. Further, it can be seen that the convective and neutral cases have heights of the area of maximum values that are around 160 m to 175 m, while the S1 and S2 cases indicate that these heights are 200 m and 225 m, respectively. This also appears to be around the same heights as the top of the surface-based temperature inversion and low-level shear layers shown



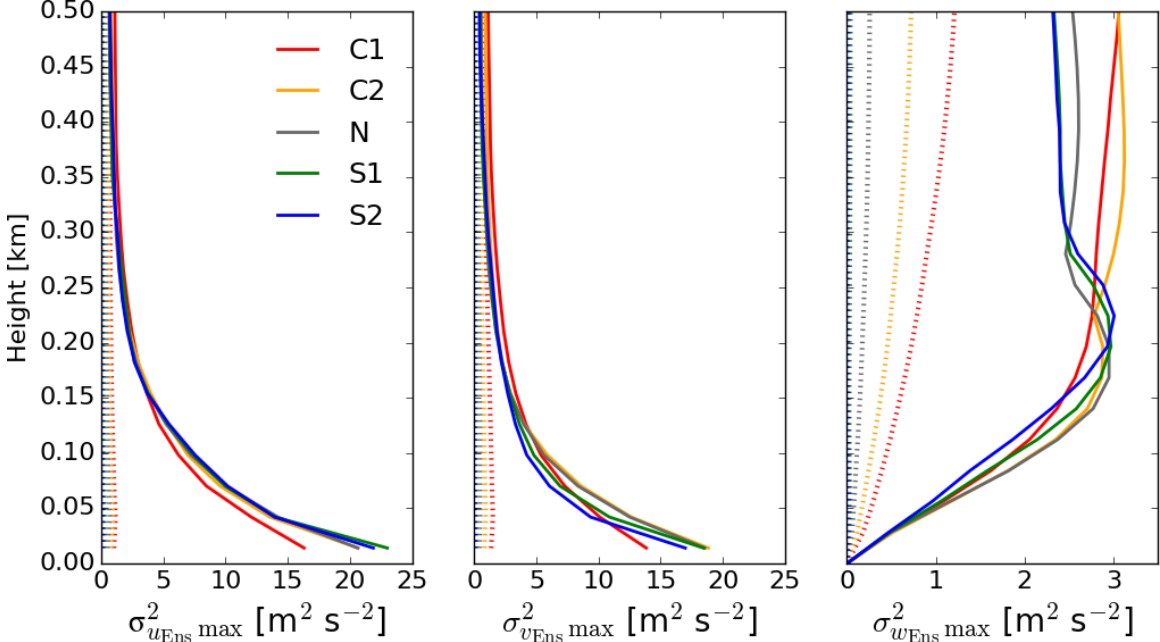

**Figure 10.** Profiles of the maximum u-, v-, and w-variance of the ensemble mean in the left, center, and right panels, respectively, for each case. In each plot, for a given height, the maximum ensemble variance is computed over a specific time interval. In the case of the environment (dotted lines), the first three minutes of the output are considered. Whereas, in the case of the downbursts, the first 13 minutes are utilized for the analysis.

in Figure 3. Thus, confirming results from Proctor (1989) where the height of the ring vortex is equivalent to that of the stable layer.

### 3.2.3 Coherent turbulent kinetic energy

Coherent, or organized, turbulent structures have been shown to increase structural loads and fatigue in wind turbines and have
5 factored into decision making for all aspects of the wind energy community from turbine design to siting and operation (Kelley et al., 2004, 2006). Kelley et al. (2004) and Kelley et al. (2006) define *coherent turbulent kinetic energy*, or CTKE, as a fluid dynamic parameter which describes both spatially and temporally organized motions in a turbulent field. While the primary focus for calculating CTKE has been for low-level jets and Kelvin-Helmholtz waves, here we compute the parameter from the organized turbulent motions of the downburst, and associated ring vortex. The calculation of CTKE as defined in Kelley et al.
10 (2004) is as follows:

$$\text{CTKE} = 0.5\sqrt{\langle u'w'\rangle^2 + \langle u'v'\rangle^2 + \langle v'w'\rangle^2} \tag{4}$$

where $\langle u'w'\rangle$, $\langle u'v'\rangle$, and $\langle v'w'\rangle$ are the domain averages of the kinematic momentum flux components. As an extension to the more conventional turbulent kinetic energy (TKE), CTKE effectively represents the amount of fluxes that are associated with



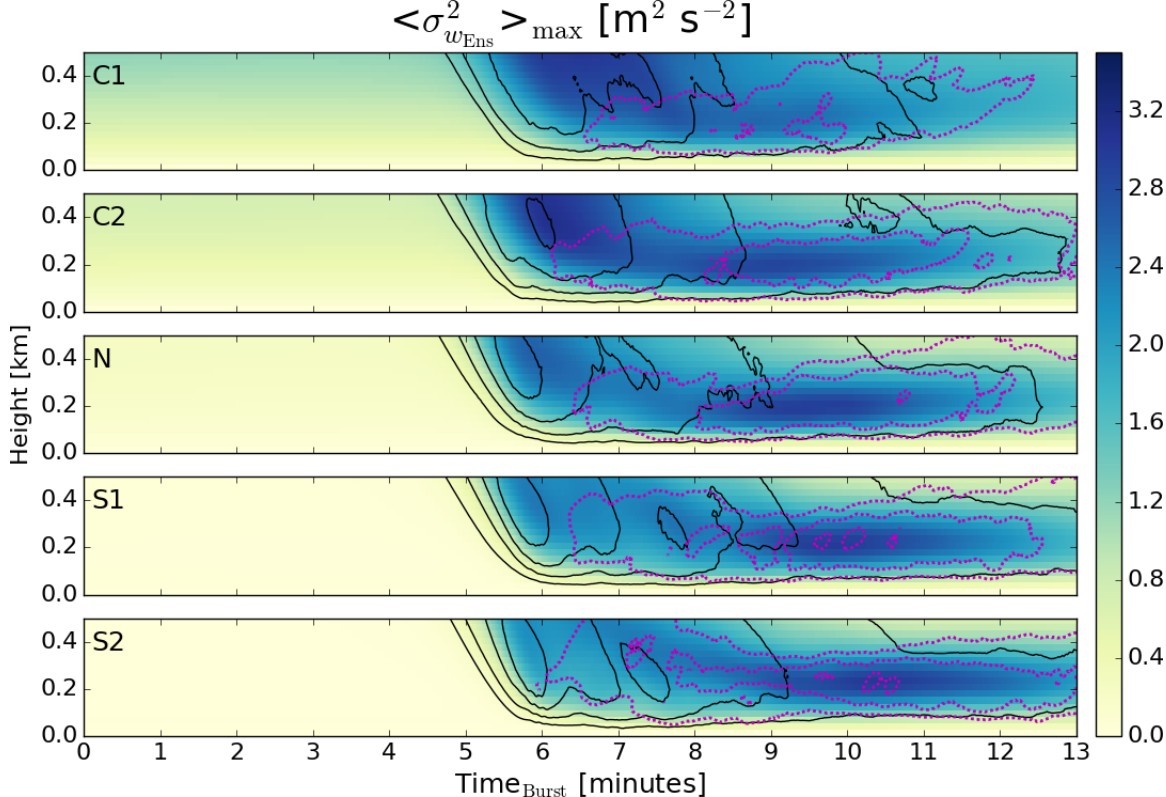

**Figure 11.** The same as Figure 7 but for vertical velocity variance.

organized structures. For example, in a completely random (homogeneous and isotropic) turbulent field, CTKE would be zero while TKE would be finite.

    The total fluxes necessary to compute CTKE are output as domain averaged profiles every second of the simulation. Figure 12 shows the now familiar time-height plots with positive and negative vertical velocity contours for the ensemble-averaged value

5  of CTKE for each case. Here we see that the convective cases generate the highest values of CTKE just as the downburst forms and strengthens. A trace of higher values of CTKE can also be seen at higher levels just after the maximum downdrafts occur and descending to where the ring vortex develops (most prominent in the neutral and stable cases). Interestingly, the simulations with the most well-defined ring vortices (S1 and S2) show the lowest values of CTKE at the levels in which the ring vortex exists. In each of the cases, the levels with the largest amounts of CTKE are in the lowest 150 m. The convective

10  cases have large amounts of CTKE through deeper levels; in contrast, case S1 shows a large amounts of CTKE focused below 50 m and decreasing quickly above.



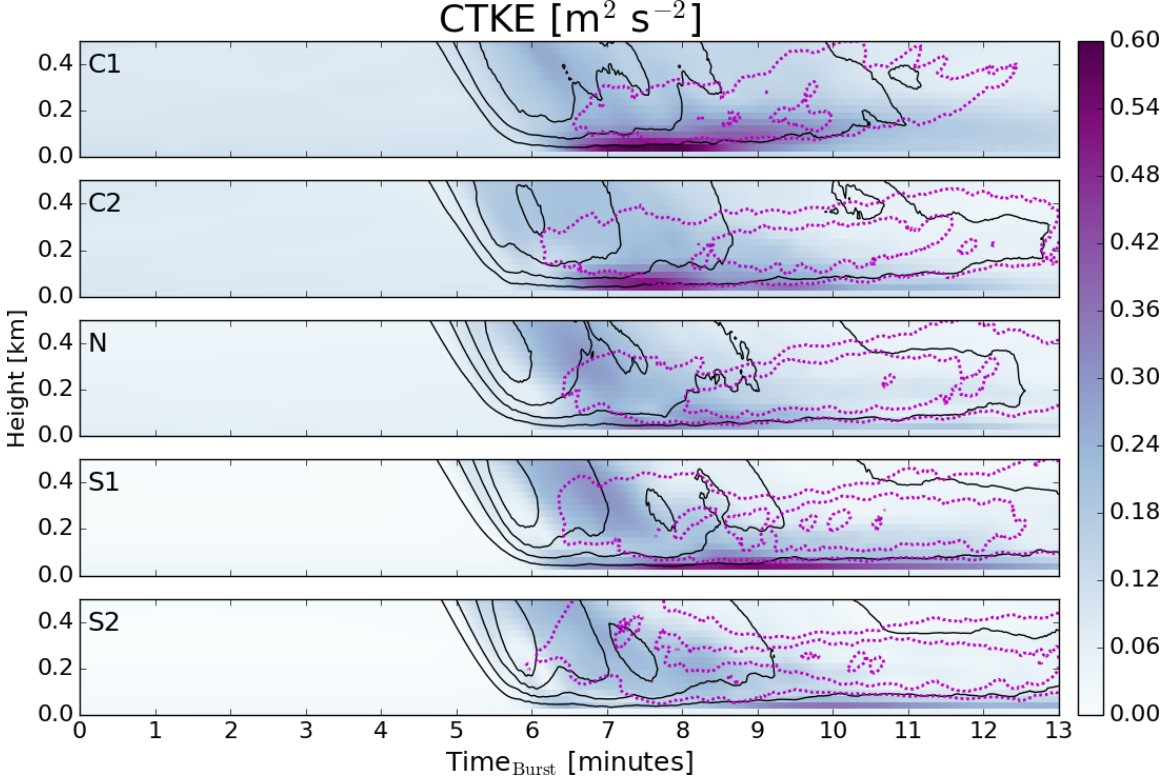

**Figure 12.** The same as Figure 7 but for coherent turbulent kinetic energy (CTKE).

### 3.2.4 Wavelet spectral analysis

In order to quantify the energy at different scales, wavelet spectral analysis of the u-component wind field at 98.13 m is performed for each ensemble member of each case. The wavelet spectrum is computed from the time series data output every 1 s for each y-location using the Daubechies 5 (db5) wavelet. Sensitivity to the wavelet was performed with the Symmlet 8 wavelet and the results showed very little dependence on the wavelet type (not shown). Due to the curvature of the outflow winds, the wind ramp does not occur for the selected vertical slice at the same time for all y. Thus, the wavelet spectra are averaged along y between 4.5 km and 5.5 km where the outflow reaches the vertical slice at roughly the same time. Further, at these points, as the outflow is almost directly to the East of the downburst center, the wind field is largely dominated by the u-component. Figure 13 shows the resulting ensemble average of the wavelet power spectra for each case. The white portions of the figures are the areas inside of the *cone of influence*; the region in which edge effects become important (Torrence and Compo, 1998). The non-stationary character of the turbulence associated with the u-component wind field is clear to see in each case. As would be expected, the ambient energy (from $\text{Time}_{\text{Burst}} = 2\text{–}6$ min) is higher for the convective cases and lower for the stable cases. In general, the lowest frequencies or small wavenumbers contain the peak in the wavelet spectra. At the time of the first impact of downbursts, energy increases by a factor of ten or more across all the scales almost instantly. After this time,





energy slowly declines across all frequencies. The convective cases resolve the highest amounts of energy at low frequencies compared to the other cases. As stability increases, the amount of energy resolved generally decreases, however, case S1 produces a fair amount more energy than the N and S2 cases. This same pattern is seen when studying CTKE (Figure 12).

## 4   Summary and conclusions

In this study, a pseudo-spectral LES code is utilized to simulate several idealized downbursts during the evening transition (ET). The ET is first simulated by spinning up a convective boundary layer and then linearly decreasing the surface temperature such that the boundary layer passes through a neutral regime and ultimately achieves a stable regime. The idealized downbursts are generated through a three-dimensional cooling source located over the top of the boundary layer to create a pocket of negatively buoyant air mimicking the latent cooling of evaporation, melting, and sublimation during a downburst. Downburst simulations

are initialized separately within the various PBL regimes, allowing for the effects of stability to be analyzed. Additionally, several instances are run for each stability regime by modifying the initial conditions in order to yield more generalized results. This is necessary due to the random locations of thermals in the convective boundary layer (and to a lesser extent, the neutral and stable boundary layers) which causes significant variation in the downburst winds. Analyses of these results allow for the following conclusions to be drawn:

– As seen with observed thunderstorm winds (see Figure 1), the ensemble-averaged maximum wind speed remains fairly constant through each stability regime. However, the stable cases produce the most consistent and strongest outflow winds when compared to the convective and neutral cases. The consistency appears to be due to the increased homogeneity in the stable wind field and lack of ambient turbulence, while the increased severity is attributed to either a stronger ring vortex due to increased low-level shear, increased negative buoyancy generated from the vertical advection

of the surface-based stable layer by the ring vortex, or a combination of the two.

– The maximum downdraft wind appears to show a direct positive correlation with increasing stability.

– The stronger ring vortex and associated stronger winds persist for a longer duration in the stable cases than in the convective and neutral cases due to the lack of turbulence to disorganize this feature. Further, the height of the ring vortex center increases with increasing stability, up to the height of the stable layer with the depth of the downburst head

(based on temperature) becoming deeper in the stable cases.

– Maximum wind speeds are realized directly downstream from the center for the convective cases and north of the center for the stable cases; it appears that wind speeds reach their maxima where the outflow propagation direction is normal to the direction of the low-level wind shear vector. In the stable cases, the direction of the wind turns more southwesterly at the lowest levels, thus aligning the ambient wind shear vector with the northeast quadrant of the outflow winds.



**Figure 13.** Ensemble-averaged wavelet power spectrum of u-component velocity [m$^2$ s$^{-1}$] at 98.13 m using the Daubechies-5 wavelet for the C1, C2, N, S1, and S2 cases from top to bottom, respectively.





- – Calculations of CTKE show generally higher amounts of energy in the convective cases as the downburst first reaches the surface and spreads laterally. Wavelet analysis shows that this sudden increase in energy occurs across all scales as the downburst passes and then slowly begins to decline.

Future work, motivated by the present study, includes a deeper investigation into the effects of stability on the dynamics of the downburst winds. Specifically, there is interest in isolating the effects of turbulence, low-level shear, and the surface-based temperature inversion in order to assess which boundary layer characteristics have the greatest impact on the downburst flow. Further, investigations into the relationship of outflow wind intensity and depth of the stable layer are planned; studies by Proctor (1989) have shown that a deep enough stable layer can inhibit downburst winds and even prevent them from reaching the ground entirely. It is also of interest to investigate the accuracy of the current modeling framework for downbursts in various stability regimes. While it is reasonable to assume that the outflow winds from downbursts may encounter various stability regimes while advancing away from the downburst center, it is unclear to what extent the environment directly underneath the downburst is turbulent during the descent. Thus, future LES studies of the downburst-producing thunderstorm utilizing microphysics parameterizations are necessary to increase the realism of the simulations and to determine the representativeness of these findings.

*Competing interests.* No competing interests are present.

*Acknowledgements.* This work was supported by the National Science Foundation (NSF) under Grant Nos. CBET-1336760 and CBET-1336304.





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
