# Peer review of "Investigating the impact of atmospheric stability on thunderstorm outflow winds and turbulence"

_Wind Energy Science, 2017_

## Referee Comment (RC1) · Anonymous Referee #1 · 9 Nov 2017

In this work, the role of the evening transition environment upon idealized downburst winds is explored. The evening transition environment is spun up and at different times during this transition, downbursts are initiated utilizing a cooling source forcing. The downburst outflow characteristics are then explored. One of the most significant findings of this work is that the strongest near-surface donwdraft winds (and subsequent maximum near-surface horizontal winds that result following the impingement of the cold downdraft) increase for increasing stability, a somewhat unintuitive result. The authors conclude that rising buoyant thermals in the less stable environment act against the downdraft, weakening it, even though the negatively buoyant air is descending in an absolutely unstable environment. The authors conclude it is the (a) suppression of turbulence in the more stable regimes (b) a stronger ring vortex (c) increased negative

buoyancy "generated from the vertical advection of the surface based stable layer by the ring vortex" that are primarily responsible for the increased winds. They also invoke RKW theory to explain the stronger roll vortex in the stable environments.

I appreciate the efforts the authors have gone through to create the evening transition environments in which the downbursts are initiated. Most, if not all, studies of this nature tend to force the downbursts in horizontally homogeneous environments. I also appreciate that the authors have run ensembles for each environment in order that some statistics can be generated.

Issues, roughly in order of importance:

I have concerns with the experimental design. In the S1 and S2 environments, cooling sources are initiated in environments where buoyant thermals are numerous. Ostensibly these buoyant thermals are forced by a surface that is warmer than the atmosphere immediately above it, from solar insolation. However, I am not convinced that there should still be enough buoyant thermals occurring underneath a real cumulonimbus cloud that is in the process of creating a dry downburst. At this stage in a thunderstorm's life, wouldn't one assume that free convection has been suppressed already by the lack of solar insolation? Or is there enough "stored heat" in the ground to continue to force convection in real storms? These thermals are the main factor the authors invoke to explain their unintuitive results, and I am wondering, is this a physically plausible experimental design? Because to me this is probably the most important result if it's right, and suggests that future simulations of downbursts should incorporate these fluxes rather than just assume a horizontally homogeneous base state.

The model grid is isotropic with a grid spacing of 28 meters. This seems to be pretty coarse vertical resolution near the ground for a study such as this that concerns wind turbines. In Figure 5, the strongest horizontal winds associated with the downburst are always found at the model's lowest grid level. With a semi-slip simulation (in this case, a surface roughness of 10 cm) I would expect there the height at which the maximum

outflow occurs to be above this level. A lot of attention in this paper is paid to the bottom 100 meters of the model domain, which only covers 3 or so grid levels. The difference in downdraft maxima and outflow maxima is pretty small, varying less than 10% from S2 to C1. In many wind engineering studies of downburst outflow, the height of the maximum horizontal winds is one of the important findings. In this paper, it's not discussed, and seems to be assumed to always occur at the lowest model grid level, which seems to indicate insufficient vertical resolution near the ground.

It appears the authors are holding the cooling source function fixed in space even though there are environmental winds at the level where the downburst is initiated. Wouldn't one expect the air-mass thunderstorm creating a downburst to be (roughly) translated to some extent with the environmental winds? Is it physically realistic to hold the cooling source fixed in space when embedded within synoptic scale winds? See for instance Orf and Anderson (1999) where their cooling source was translated in the same direction as the environmental winds in a simple linearly sheared environment.

One would expect that there would come a point where the stable layer would ultimately become deep or strong enough to result in the "intuitive" result of suppressing strong outflow winds. Have the authors run additional experiments in a later nighttime environment to see at what point that point occurs? While downbursts aren't very common late at night, they certainly do happen (and sometimes result in heatbursts!).

The authors describe the stabler environment as having more low-level shear. A hodograph of all five environments (I presume you'd need to take average winds at each level) at the time of downburst forcing would be useful to quantify this; you might be able to combine them in a single figure.

Concerning the important role of turbulence / buoyant thermals, I'd like to see a horizontal cross section of the vertical component of the wind at the time the downbursts are initiated to get a feel for the scale and strength of these thermals, that seem to be playing a crucial role in weakening the downdraft.
p17L13 "It is also speculated that with a shallow surface-based stable layer such as in this current study, it may be possible for the increased negative buoyancy generated from the lifting of the colder air at the surface to further strengthen the outflow and ring vortex in the stable cases." I do not understand this sentence, which seems to be somewhat tautological. With a shallow stable layer, one would expect adiabatic compression of the downdraft to lead to warmer (or at least less negatively buoyant) surface air as it descends through an absolutely stable layer. How does the perturbation potential temperature get *more* negative descending in a shallow stable layer? I do not understand the physics behind this sentence.

Why are the authors running with such a short time step on their finest mesh? 0.1 seconds seems to be unnecessarily small considering the maximum winds found in the simulation (Courant Friedrichs Lewy). Are the authors using a model with acoustic substepping?

In Figure 5, the vertical cross section is held fixed at 6 km. Why not move it so that it's in the same roll-vortex relative location? The top profile cross section seems especially too far west compared to the other two. It's not clear that this is the best way to compare the simulation velocity profiles.

Please be sure to create all of your line plots as vector graphics, not bitmapped, such that they can be zoomed in for the PDF version of the paper.

---

## Referee Comment (RC2) · Anonymous Referee #2 · 17 Nov 2017

The manuscript describes simulations of downbursts with a LES model and tries to identify the influence of the stabilisation of the thermal stratification of the surface layer during the evening transition on the strength of these downbursts.

The paper is well written and understandable and does not contain any obvious errors. The introduction is rather long and has got a review character which covers more than that what is necessary to understand the content of the manuscript.

The only shortcoming - to the estimation of the reviewer - is that the simulated peak wind speed of roughly 35 to 38 m/s is not compared to the vast body of literature mentioned in the introduction.

Otherwise, the manuscript may be published as is.

---

## Author Comment (AC1) · 21 Dec 2017

**Response to reviewers' comments**

We would like to thank the reviewers for their time, and their evaluations of this paper. We have carefully read these comments (shown here in grey text) and address them here (black text). The additional text to be added to the manuscript is shown in blue.

**Reviewer #1**

In this work, the role of the evening transition environment upon idealized downburst winds is explored. The evening transition environment is spun up and at different times during this transition, downbursts are initiated utilizing a cooling source forcing. The downburst outflow characteristics are then explored. One of the most significant findings of this work is that the strongest near-surface donwdraft winds (and subsequent maximum near-surface horizontal winds that result following the impingement of the cold downdraft) increase for increasing stability, a somewhat unintuitive result. The authors conclude that rising buoyant thermals in the less stable environment act against the downdraft, weakening it, even though the negatively buoyant air is descending in an absolutely unstable environment. The authors conclude it is the (a) suppression of turbulence in the more stable regimes (b) a stronger ring vortex (c) increased negative buoyancy "generated from the vertical advection of the surface based stable layer by the ring vortex" that are primarily responsible for the increased winds. They also invoke RKW theory to explain the stronger roll vortex in the stable environments. I appreciate the efforts the authors have gone through to create the evening transition environments in which the downbursts are initiated. Most, if not all, studies of this nature tend to force the downbursts in horizontally homogeneous environments. I also appreciate that the authors have run ensembles for each environment in order that some statistics can be generated.

We would like to thank the reviewer for their succinctly summarizing our work and for acknowledging the novelty of this work and analysis.

Issues, roughly in order of importance:
I have concerns with the experimental design. In the S1 and S2 environments, cooling sources are initiated in environments where buoyant thermals are numerous. Ostensibly these buoyant thermals are forced by a surface that is warmer than the atmosphere immediately above it, from solar insolation. However, I am not convinced that there should still be enough buoyant thermals occurring underneath a real cumulonimbus cloud that is in the process of creating a dry downburst. At this stage in a thunderstorm's life, wouldn't one assume that free convection has been suppressed already by the lack of solar insolation? Or is there enough "stored heat" in the ground to continue to force convection in real storms? These thermals are the main factor the authors invoke to explain their unintuitive results, and I am wondering, is this a physically plausible experimental design? Because to me this is probably the most important result if it's right, and suggests that future simulations of downbursts should incorporate these fluxes rather than just assume a horizontally homogeneous base state.

The reviewer's comment here is in essence about the impacts of cloud shading on the turbulent structure of the boundary layer. This is a fair criticism and while this topic is not well understood as

it pertains to thunderstorms and deep convection, we argue that this setup is indeed valid based on the available literature on the topic.

- To the best of our knowledge, the only convective storm study in a convective boundary layer that considered the effects of cloud shadowing came from Nowatarski et al. (2014). This study did show that cloud shadowing can reduce the boundary layer turbulence at low levels; however, the study considered a supercell thunderstorm with a large anvil cloud, which is an extreme case of cloud shadowing. The cast shadow would be very large and the time spent during which the environment is within the shadow would be longer than 100 minutes (greater than the expected lifetime of a single-cell thunderstorm).
- Given that downbursts can be generated by parent thunderstorms in short periods of time and that the decay of turbulence is not abrupt, it is reasonable to assume that turbulence may be present beneath a descending downburst. Further, downbursts are not constrained to descend directly beneath the cloud shadow and, in fact, footage of a downburst illuminated by the sun partly inspired this research. (See video below) https://www.youtube.com/watch?v=a_G2KRzha7o
- Large-eddy simulations of convective boundary layers with shallow cumulus [e.g. Schumann et al. (2002), Lohou and Patton (2014)] have shown that cloud shadows can reduce, but do not eliminate, turbulent mixing beneath clouds and that sensible heat flux is reduced (though not to zero) at low levels but, in fact, remains mostly unchanged (if not a bit strengthened) above this height. This is similar to the findings of Nieuwstadt and Brost (1986) where the evening transition was simulated by abruptly cutting off the source of heating that generated the convective boundary layer, resulting in a decay of turbulence from the bottom up.
- Lastly, we believe that one of the most important results of our work is that the outflow is greatly affected by the environmental stability. It is not only the downdraft that is reduced by the ambient turbulence but the outflow winds and lifetime of the downburst. While it may be argued that the convective cases in these idealized simulations may be oversimplified due to the presence of convection directly beneath the downdraft, a definitive argument cannot be made that the outflow will not encounter ambient convection.

In the original manuscript, we made cautionary remarks on this topic with the following text in the future work section: "[w]hile it is reasonable to assume that the outflow winds from downbursts may encounter various stability regimes while advancing away from the downburst center, it is unclear to what extent the environment directly underneath the downburst is turbulent during the descent. Thus, future LES studies of the downburst-producing thunderstorm utilizing microphysics parameterizations are necessary to increase the realism of the simulations and to determine the representativeness of these findings."

The model grid is isotropic with a grid spacing of 28 meters. This seems to be pretty coarse vertical resolution near the ground for a study such as this that concerns wind turbines. In Figure 5, the strongest horizontal winds associated with the downburst are always found at the model's lowest grid level. With a semi-slip simulation (in this case, a surface roughness of 10 cm) I would expect there the height at which the maximum outflow occurs to be above this level. A lot of

attention in this paper is paid to the bottom 100 meters of the model domain, which only covers 3 or so grid levels. The difference in downdraft maxima and outflow maxima is pretty small, varying less than 10% from S2 to C1. In many wind engineering studies of downburst outflow, the height of the maximum horizontal winds is one of the important findings. In this paper, it's not discussed, and seems to be assumed to always occur at the lowest model grid level, which seems to indicate insufficient vertical resolution near the ground.

We would like to thank the reviewer for this comment on the vertical resolution within these simulations as it pertains to the height of the downburst jet. Considering past studies that have solved the equations of motions of downburst winds utilizing LES or high-resolution RANS approaches, the following table describes the height of the resolved jet as well as the vertical grid spacing and roughness length considered in the present study.

| Journal Article | z [m] | $z_i$ [m] | Jet height [m] |
|---|---|---|---|
| Present study | 28 | 0.1 | 28 (lowest level) |
| Anabor et al. (2011) | 15.5 | 0.1 | 15.5 (lowest level) |
| Mason et al. (2009)[*+] | 1 - 20 m (stretched) | 0.2 | 11 |
| Mason et al. (2010)[+] | 1 - 20 m (stretched) | 0.2 | 10-15 |
| Vermeire et al. (2011a) | 1 - 50 m (stretched) | 0.1 | 15 |
| Vermeire et al. (2011b) | 1 - 50 m (stretched) | 0.01 | 15 |
| Oreskovic (2016) | 1 - 50 m (stretched) | 0.1 | Around 8 m |

[*] 2-dimensional study; [+] URANS model

It can be seen that the height of the jet in each of the cited studies is very close to the surface and lower than the height of the first model level in the current study. Further, these heights are not uniform for a constant roughness length; they are dependent on physical properties (such as the cooling source intensity) and the computational approach (URANS or LES). In the present study, consistent with the selected vertical resolution, the height of the jet is not the focus of any analysis (which is rather found to generally occur below the turbine hub height). It should also be noted that studies that are able to resolve the jet profile (i.e., where the strongest wind speed is not at the lowest model level) are either two-dimensional (and, thus, not valid for the current study) or utilize a stretched vertical grid.

At this point, we would like to point out that we have utilized a dynamic (tuning-free) technique in conjunction with the so-called Smagorinsky subgrid-scale (SGS) model. By construction, this SGS model assumes an isotropic length-scale. In order to perform LES runs with a severely stretched vertical grid (i.e., with a large aspect ratio of horizontal to vertical resolution), one must adopt an anisotropic SGS model [e.g., Scotti and Meneveau (1993)]. Since we do not have such an SGS model in our code, we attempted to maintain as close to an isotropic grid as possible.

Interestingly, most of the published microburst modeling studies using stretched vertical grids have not made use of any anisotropic SGS models; in our view, results from these simulations are questionable.

It appears the authors are holding the cooling source function fixed in space even though there are environmental winds at the level where the downburst is initiated. Wouldn't one expect the air-mass thunderstorm creating a downburst to be (roughly) translated to some extent with the environmental winds? Is it physically realistic to hold the cooling source fixed in space when embedded within synoptic scale winds? See for instance Orf and Anderson (1999) where their cooling source was translated in the same direction as the environmental winds in a simple linearly sheared environment.

Similar simulations with a moving source have, indeed, been carried out and compared to those with a stationary source (see for example, Mason et al. 2010). In Mason et al. (2010), it was shown that the "tilted" downburst (which is a stationary source in an environment with a non-zero geostrophic wind, identical to the simulations in the current study) and "translating" downburst (as the reviewer has mentioned) show several similar characteristics and are both observed in nature. While both approaches are valid, given that many of the downburst studies in the literature utilize the stationary downburst (for example, Anabor et al. 2011, Anderson et al. 1992, Mason et al. 2009 [both papers], Oreskovic 2016, Vermeire et al. 2011 [both papers]), we felt it was appropriate to also employ this approach.
The following has been added to the revised manuscript on page 8, line 16:
"The cooling source is treated as stationary as is quite common in the downburst literature (Anderson et al. 1992, Mason et al. 2009a, Mason et al. 2009b, Anabor et al. 2011, Vermeire et al. 2011a, Vermeire et al. 2011b, Oreskovic 2016). Mason et al. (2009b) compared this method with one using a translating source and concluded that both are viable methods for simulating downbursts."

One would expect that there would come a point where the stable layer would ultimately become deep or strong enough to result in the "intuitive" result of suppressing strong outflow winds. Have the authors run additional experiments in a later nighttime environment to see at what point that point occurs? While downbursts aren't very common late at night, they certainly do happen (and sometimes result in heatbursts!)

We are very pleased to see this comment. Additional simulations have, in fact, been carried out with the goal of simulating heat bursts and the theoretical "mid-air microburst." From these additional simulations, it appears that downburst strength begins to weaken slightly in the hour after the S2 simulation (see S3 and S4 in Figure 1 below). These simulations are investigative in nature and, thus, an ensemble was not generated. As such the results cannot be deemed definitive; however, it does appear that eventually the stable layer will act to inhibit the downburst strength as intuition would suggest. We are reluctant to include these findings in the final manuscript due to the limited resolution employed while simulating a deep stable layer. In order to faithfully simulate such a (very) stable surface layer, a resolution on the order of 1 m would be required.

That said, heat burst signatures (high outflow wind speeds associated with warm temperature perturbations) were detected even in the S2 simulations of the present study (see Figure 2 below); however, we believe that the additional analyses and simulations are not within the scope of the current work. It is quite possible that such analyses may be the focus of a future study.

The following sentence has been added in the future work section on page 26, line 9:

"These studies will also allow possible relationships to be established between the depth of the stable layer and the strength of a downburst in producing a *heat burst*."

[Figure]

**Figure 1 - Variations of U$_{storm}$ (top), w$_{min}$ (middle), and the ratio between the two (bottom) for each case - C1 (red), C2 (yellow), N (grey), S1 (green), and S2 (blue) - with the average bulk Richardson number at the first model level over the first three minutes of simulation. Circles represent the 0x_0y cases, right-pointing triangles represent the 5x_0y cases, upward-pointing triangles represent the 0x_5y cases, and diamonds represent the 5x_5y cases. The ensemble mean is denoted by an "x" for each case. For the S3 and S4 cases, only one simulation (0x_0y) was run; thus, they are denoted only by an "x" in this plot.**

[Figure]

**Figure 2 - 2D histogram of wind speed versus potential temperature perturbation at the lowest level in the 0x_0y simulations for S1 (top), S2 (second row), S3 (third row), and S4 (bottom row). The time is broken into segments with outflow from 5 to 8 minutes in the first column, from 8 to 11 minutes in the middle column, and from 11 to 13 minutes in the right column.**

The authors describe the stabler environment as having more low-level shear. A hodograph of all five environments (I presume you'd need to take average winds at each level) at the time of downburst forcing would be useful to quantify this; you might be able to combine them in a single figure.

We agree that this may have been under-emphasized in the average profile figures in the manuscript's Figure 3. Please see our response to the following comment to see how this is addressed.

Concerning the important role of turbulence / buoyant thermals, I'd like to see a horizontal cross section of the vertical component of the wind at the time the downbursts are initiated to get a feel for the scale and strength of these thermals, that seem to be playing a crucial role in weakening the downdraft.

We have generated a new figure to show the strength of the buoyant thermals (w-component wind speed) within a single member of the C1, N, and S2 simulations at the onset of the downburst simulation. In the horizontal cross sections (Figure 3), the general scale of the thermals in the x-y plane can be seen; however, the vertical scale of the thermals is understated. By studying vertical profiles (Figure 4), the horizontal extent can be deduced, and the vertical

extent and height dependence are made much clearer. To this plot, as the previous comment of the reviewer also suggests, we have added panel plots on the right indicating the contributions of the u- and v-component wind speeds in these simulations.

[Figure]

**Figure 3 - Horizontal cross sections of vertical wind speeds at two heights, 0.196 km (left) and 0.981 km (right) for the C1-5x_5y (top), N-5x_0y (middle), and S2-0x_5y (bottom) cases.**

[Figure]

**Figure 4 - Vertical cross sections along the East-West axis at the y-location where the maximum vertical velocity occurs for the C1-5x_5y (top row), N-5x_0y (middle row), and S2-0x_5y (bottom row) cases at the start of the downburst runs. Filled contours show vertical velocity [m s$^{-1}$ ] with arrows representing the u-w wind field. In the right column are the domain-averaged u- and v-component wind speed profiles in black and red, respectively.**

We appreciate the reviewer's comment and have added Figure 4 to our revised manuscript along with the following text:

Figure 4 shows a vertical cross section of the w-component velocity field for the C1--5x_5y (top row), N--5x_0y (middle row), and S2--0x_5y (bottom row) cases along with average profiles of the u- and v-component winds in the right column. These cross-sections are taken from the location of maximum vertical velocity in order to get a sense for the strongest of the updrafts within the domain. As can be seen, in the C1--5x_5y simulation, strong thermals are present within the model domain in which vertical velocities exceed 6 m s$^{-1}$. As the surface heating decreases, the strength of the thermals becomes weaker. However, by the time the atmosphere reaches a near-neutral regime, remnants of these thermals are still prevalent throughout the domain. After two additional hours of stabilization, these remnants have largely dissipated as shown in the S2--0x_5y panel.

Upon analysis of the average profiles of the u- and v-component winds in Figure 4, it is clear that there is an increase in v-component wind speeds as stability increases. In the stable cases, a

sharp increase in the v-component winds below roughly 100 m is seen due to the Coriolis force causing winds to veer with height within the SBL (Stull 2012)."

p17L13 "It is also speculated that with a shallow surface-based stable layer such as in this current study, it may be possible for the increased negative buoyancy generated from the lifting of the colder air at the surface to further strengthen the outflow and ring vortex in the stable cases." I do not understand this sentence, which seems to be somewhat tautological. With a shallow stable layer, one would expect adiabatic compression of the downdraft to lead to warmer (or at least less negatively buoyant) surface air as it descends through an absolutely stable layer. How does the perturbation potential temperature get *more* negative descending in a shallow stable layer? I do not understand the physics behind this sentence.

We thank the reviewer for pointing out the confusion in this sentence. To avoid any potential confusion, this sentence has been removed from the revised manuscript.

Why are the authors running with such a short time step on their finest mesh? 0.1 seconds seems to be unnecessarily small considering the maximum winds found in the simulation (Courant Friedrichs Lewy). Are the authors using a model with acoustic substepping?

This time step is necessary for a follow-up study underway where the outflow is used in wind turbine aeroelastic response simulation within the FAST (Fatigue, Aerodynamics, Structures, and Turbulence) model in order to calculate loads and stresses resulting from such flow fields on a wind turbine.

The following sentence has been added to the manuscript at page 7, line 13:
"This decrease in time step is applied in order to ensure numerical stability during the computation of wind turbine loads derived from aeroelastic response simulations using the Fatigue, Aerodynamics, Structures, and Turbulence (FAST) open-source simulation tool."

In Figure 5, the vertical cross section is held fixed at 6 km. Why not move it so that it's in the same roll-vortex relative location? The top profile cross section seems especially too far west compared to the other two. It's not clear that this is the best way to compare the simulation velocity profiles.

We agree that this was not the ideal method to compare the profiles. Initially, more analysis was planned for the vertical slice at x = 6 km which is what led to selection of the profiles at this location. Since most of that analysis was subsequently removed, we agree that the slice should be located at a similar location between runs. Thus, we have adjusted the analysis so as to be at the location of the maximum wind speed for each case in order to keep the analysis consistent with the flow fields in a relative sense.

Figure 5 has been updated in the manuscript.

Please be sure to create all of your line plots as vector graphics, not bitmapped, such that they can be zoomed in for the PDF version of the paper.

Thank you very much for pointing this out. We have converted all .png images to .eps format in the revised manuscript.

---

## Author Comment (AC2) · 21 Dec 2017

**Response to reviewers' comments**

We would like to thank the reviewers for their time, and their evaluations of this paper. We have carefully read these comments (shown here in grey text) and address them here (black text). The additional text to be added to the manuscript is shown in blue.

**Reviewer #2**

The manuscript describes simulations of downbursts with a LES model and tries to identify the influence of the stabilisation of the thermal stratification of the surface layer during the evening transition on the strength of these downbursts.

The paper is well written and understandable and does not contain any obvious errors. The introduction is rather long and has got a review character which covers more than that what is necessary to understand the content of the manuscript.

We thank the reviewer for these comments and for the review of our paper.

The only shortcoming - to the estimation of the reviewer - is that the simulated peak wind speed of roughly 35 to 38 m/s is not compared to the vast body of literature mentioned in the introduction.

We agree that the maximum simulated wind speed of 35 to 38 m/s ought to have been compared with other simulation studies as well as with observed downbursts in order to bolster the validity of these simulations.

To emphasize this, the following sentence has been added to the revised manuscript:

"The maximum wind speeds of 35--38 m/s are consistent with strong observed downbursts: 25--30 m/s in Wakimoto (1985); 32 m/s (with theorized maxima in the F3 range of 70-92 m/s) in Fujita (1981); and 67 m/s in Fujita (1985). Furthermore, other simulation studies have reported similar wind speed maxima of 38 m/s (Orf et al. 2012), 35 m/s (Anabor et al. 2011), 57 m/s (Mason et al. 2009), 35--65 m/s (Oresekovic 2016), 24--32 m/s (Orf and Anderson 1998), and 47 m/s (Vermeire et al. 2011)."

Otherwise, the manuscript may be published as is.

We thank the reviewer for these comments and for recommending publication of this work.